# Effect of statins on inflammation and cardiac function in patients with chronic Chagas disease: A protocol for pathophysiological studies in a multicenter, placebo-controlled, proof-of-concept phase II trial

**Carolina Campos-Estrada[1], Edurne Urarte[2], Marisol Denegri[2], Litzi Villalón[3], Fabiola González-Herrera[4], Ulrike Kemmerling[5]*, Juan D. Maya[4]\***

**1** Facultad de Farmacia, Universidad de Valparaíso, Valparaíso, Chile, **2** Departamento de Pediatría y Cirugía Infantil Occidente, Facultad de Medicina, Universidad de Chile, Santiago, Chile, **3** Servicio de Salud Viña del Mar-Quillota, Hospital Gustavo Fricke, Viña del Mar, Chile, **4** Programa de Farmacología Molecular y Clínica, Instituto de Ciencias Biomédicas, Facultad de Medicina, Universidad de Chile, Santiago, Chile, **5** Programa de Biología Integrativa, Instituto de Ciencias Biomédicas, Facultad de Medicina, Universidad de Chile, Santiago, Chile

* jmaya@med.uchile.cl (JDM); ukemmerling@uchile.cl (UK)

**Data Availability Statement:** No datasets were generated or analysed during the current study. All

## Abstract

### Background

Cardiac complications, including heart failure and arrhythmias, are the leading causes of disability and death in Chagas disease (CD). CD, caused by the *Trypanosoma cruzi* parasite, afflicts 7 million people in Latin America, and its incidence is increasing in non-endemic countries due to migration. The cardiac involvement is explained by parasite-dependent, immune-mediated myocardial injury, microvascular abnormalities, and ischemia. Current treatment of early CD includes the administration of nifurtimox and benznidazole. However, their efficacy is low in the chronic phase and may induce severe adverse events, forcing therapy to halt. Therefore, finding innovative approaches to treat this life-threatening tropical disease is of utmost importance. Thus, improving the efficacy of the current antichagasic drugs by modifying the inflammatory response would render the current treatment more effective. It has been reported that, in mice, simvastatin decreases cardiac inflammation and endothelial activation, and improves cardiac function, effects that require clinical confirmation.

### Objective

The study aims to analyze whether two doses of Atorvastatin, administered after CD treatment is completed, are safe and more efficacious than the antiparasitic drugs alone in reducing general inflammation and improving endothelial and cardiac functions in a proof-of-concept, placebo-controlled phase II trial.

relevant data from this study will be made available upon study completion.

**Funding:** CC: 11201312 Agencia Nacional de Investigacion - FONDECYT iniciación - Chile JDM, 1210359 Agencia Nacional de Investigacion - FONDECYT regular - Chile UK: 1220105, Agencia Nacional de Investigacion - FONDECYT regular - Chile FG: 21170427, Agencia Nacional de Investigacion - Becas Doctorado - Chile The funders had and will not have a role in study design, data collection and analysis, decision to publish, or preparation of the manuscript.

**Competing interests:** The authors have declared that no competing interests exist.

## Methods

300 subjects will be recruited from four Chilean hospitals with an active Program for the Control of Chagas Disease. 40 or 80 mg/day of atorvastatin or placebo will be administered after completion of the antichagasic therapy. The patients will be followed up for 12 months. Efficacy will be determined by measuring changes in plasma levels of anti-inflammatory and pro-inflammatory cytokines, soluble cell adhesion molecules, BNP, and cTnT. Also, the resting 12-lead ECG and a 2D-echocardiogram will be obtained to evaluate cardiac function.

## Trial registration

ClinicalTrials.gov Identifier: NCT04984616.

## Introduction

Chagas Disease is an illness affecting about 7 million people in Latin America and is caused by the *Trypanosoma cruzi* parasite. That is a zoonosis transmitted by a hematophagous insect of the Triatominae subfamily [1]. As CD can also be transmitted by infected blood or congenitally, migration contributes to spreading to non-endemic regions [2]. CD is the leading cause of death from parasitic diseases, even above malaria [3, 4]. Preventive measures include vector control, blood donor screening, congenital transmission detection, and treating infected children and acute cases [2]. Nonetheless, the effectiveness of current control programs and drug therapy is limited [2], explaining why the 2020 WHO plans to eradicate CD were not achieved [5].

CD evolves in two phases, acute and chronic. The acute phase is asymptomatic or with mild, self-limited, nonspecific symptoms [6]. The chronic phase may remain symptomless but with positive serology, although a mild inflammatory state may persist, so it is also called the indeterminate phase [7]. After 10 to 30 years, up to one-third of patients may present symptoms due to damage to the heart, esophagus, or colon. Chronic Chagasic Cardiomyopathy (CCC) provokes heart failure [8], arrhythmias [9], or thromboembolic phenomena and can cause sudden death [10].

The clinical diagnosis of CD in the acute phase is difficult because of its asymptomatic feature. However, detecting the parasite in the patient's blood is possible by identifying the trypomastigotes in the buffy coat or by a conventional polymerase chain reaction (PCR). In the chronic phase, the diagnosis relies on clinical and epidemiological data and serologic tests such as a highly sensitive IgG ELISA and indirect IgG immunofluorescence [1, 11]. Abdominal radiological studies should be conducted if there is suspicion of gastrointestinal involvement. Cardiac evaluation is essential when the diagnosis is made to provide a functional staging and to stratify a ten-year mortality risk [12–14]. Functional staging aids in explaining the evolution of CD. It is based on clinical, the NYHA functional classification, X-Ray, electrocardiogram (ECG), echocardiogram, 24-hour Holter, thromboembolism, and sudden death criteria, and describes stages I to IV [14].

Different biomarkers have been proposed to evaluate progression, prognosis, or response to treatment. Still, none has demonstrated sufficient specificity to be incorporated as a gold standard for diagnosing Chagas disease [15]. However, brain natriuretic peptide (BNP) and cardiac troponin T (cTnT) may help predict progress towards left ventricular dysfunction [16]. Parasite DNA detection is a valuable marker only for determining treatment failure since a negative parasitemia does not exclude the persistence of the parasite [17, 18].

Three pathogenic mechanisms explain CCC. **i)** A parasite-dependent, immune-mediated myocardial damage, which is the most critical determinant of the disease [19], where the $T_H1/T_H2/T_{Reg}$ response is a crucial feature for parasite evasion because the immune response polarizes toward a broad diversity of pro- (IFN-γ, TNF-α, IL-1β) and anti-inflammatory (IL-4, IL-10, IL-17A) profiles [20–23]. **ii)** Microvascular abnormalities and ischemia [24] that may be related to regional perfusion disturbances reverted by dipyridamole [25, 26], platelet activation [27], and endothelial dysfunction, as evidenced by an increase in the cell adhesion molecules Intercellular Adhesion Molecule type 1 (ICAM-1), Vascular Cell Adhesion Molecule (VCAM) and E-selectin, including a soluble form of the cell adhesion molecules (sCAM) [28–30], as seen in hearts of chronically *T. cruzi*-infected mice, which are similar to that observed in the ischemic cardiomyopathies [24]. And **iii)** Cardiac dysautonomia secondary to self-reactive immune mechanisms [31, 32].

Thus, the IFN-γ, IL-1β, IL-4, IL-17A, and IL-10 profiles might help to predict the progression of CD and its outcome after drug therapy [33], and the inclusion of anti-inflammatory drugs might favorably incline toward a better immune response profile and improve the treatment of CCC. Moreover, myocardial perfusion disorders, due to endothelial damage and microcirculatory alterations, contribute to the progress of the segmental LV dysfunction observed in the chronic phase of CCC [34]. Thus, any therapeutic approach to improving left ventricle function or preventing these vascular derangements could benefit patients with CD. Therefore, endothelial adhesion molecules are potential biomarkers for evaluating anti-inflammatory drugs' impact on preventing the progression of LV dysfunction [15]. E-selectin may be an excellent candidate since it is expressed exclusively by vascular EC [35].

The etiologic treatment of CD is done with 5–10 mg/kg/day nifurtimox (NFX) or 5 mg/kg/day benznidazole (BZD) for 60 days [1]. All patients should receive antichagasic treatment without exclusion or delay, except if there is any contraindication such as hypersensitivity reactions, severe renal, liver, or cardiac disease, including CCC with structural involvement, alcoholism (Antabuse effect of NFX), and severe megacolon or megaesophagus [1]. The adverse events produced by these two drugs can force the suspension of the therapy in up to 48% of patients receiving NFX [36–38].

Drug therapy during the acute and early indeterminate phases is considered curative [39]. However, it is more difficult to declare a cure for chronic infection, and their efficacy is weak or controversial, especially when mortality is considered [40, 41]. Moreover, a long-term follow-up is required to confirm seroreversion, making serology an inadequate surrogate of cure in chronic CD [42]. Novel therapies, like the antifungals posaconazole or ravuconazole, were unsuccessful [43–45]. Therefore, it is of utmost importance to find innovative approaches to treat this neglected life-threatening tropical disease, including modifying the host's inflammatory response–primarily by repurposing existent drugs–.

The specialized pro-resolving molecules involved in the natural resolution of inflammation include several lipids derived from essential fatty acids of the plasma membrane that control the magnitude and duration of local inflammation [46]. Interestingly, aspirin and the cholesterol-lowering statins, including atorvastatin (ATO), switch the cyclooxygenase-2 (COX2) activity to produce precursors, no longer for the synthesis of prostaglandins (which are pro-inflammatory) but epimers that, through the action of 5-lipooxygenase (5-LO), will result in more stable pro-resolutory molecules, such as 15-epi-lipoxin $A_4$ (LXA$_4$) or aspirin-triggered resolvin D1 (AT-RvD1) [47, 48]. Thus, aspirin and statins may decrease cardiac inflammation and endothelial activation in chronically *T. cruzi*-infected mice [49–51], an action mediated by 15-epi-LXA4 [28–30]. Furthermore, simvastatin improves, independently of 15-epi-LXA$_4$, the ventricular function in chronically infected BALB/c mice [52], while AT-RvD1 improves cardiac electrical function in the absence of trypanocidal treatment [53]. Although the preclinical

evidence suggests that a statin could benefit chronic CD, this issue has not been proven in the clinical setting.

ATO's therapeutic and safety profiles are well known, as are their mechanism of action and pharmacological actions, including their anti-inflammatory properties, which are shared by the other members of the statin class. Importantly, due to the high efficacy and low incidence of severe AE, it is one of the most widely used statins today [54]. 10–80 mg/day of ATO is used to decrease the so-called LDL cholesterol involved in the pathogenesis of atherosclerotic cardiovascular disease.

Statins are not currently approved for CD treatment. Consequently, it is proposed to analyze whether two different doses of ATO, after antiparasitic therapy (NFX or BZD) is completed, are safe and more efficacious than the antiparasitic therapy alone in reducing general inflammation and in improving the endothelial and cardiac functions in a proof-of-concept, phase II clinical trial.

The specific aims include:

1. To evaluate the efficacy of the combination of ATO and antichagasic therapy to decrease inflammation, as measured by the plasma levels of the cytokines, TNF-α, IFN-γ, IL-10, IL-1B, IL-4, and IL-17A; endothelial activation, as measured by the plasma levels of sCAM: sE-selectin, sICAM-1, and sVCAM-1; and cardiac damage and function, as measured by the plasma levels of BNP, and cTnT, the resting 12-lead ECG (cardiac rate, QT segment duration, changes in electrical conduction as determined by QRS segment duration and morphology), and ejection fraction determined by a 2D-echocardiogram.

2. To determine the safety and tolerability of the combination of ATO with antichagasic therapy, as measured by the incidence of AE (e.g., rhabdomyolysis) and discontinuation of treatment.

3. To evaluate the response to treatment of the combination of ATO and antichagasic therapy, measured by quantitative PCR and serology, during a follow-up period of ten months.

4. To determine drug compliance for the therapies by measuring drug accountability throughout the study.

## Methods

A randomized, placebo-controlled, multicentered, phase II clinical trial will be performed, with three parallel arms and double blinding. The present protocol follows the SPIRIT guidelines [55]. The study was registered with ClinicalTrials.gov (registration number: NCT04984616).

### Study design

**Outcomes.** The twelfth-month primary outcome of this study is the change in cardiac function in patients with chronic CD. The secondary outcome is settled at the change in plasma levels of biomarkers versus changes in parasite load.

The number of patients with a change in the stage of chronic cardiomyopathy within 12 months of starting antichagasic therapy will be considered to assess the primary endpoint.

We will evaluate whether the effect of atorvastatin after antiparasitic therapy (NFX or BZD) is more effective than antiparasitic therapy alone (placebo group) in preventing the onset of cardiac disorders determined by non-progression from phase A according to the I Latin American Guidelines for the diagnosis and treatment of chagasic cardiomyopathy [56]. To make this assessment, significant changes in i) electrocardiogram (heart rate and QT interval duration, as well as the appearance of electrical conduction disorders, determined by the

duration and morphology of the QRS segment), ii) ejection fraction (assessed by echocardiography), iii) cardiac silhouette size (chest X-ray) will be considered.

Additional measures of scientific value include (a) the change in the plasma levels of the biomarkers of inflammation and endothelial activation; (b) the incidence and severity of AEs; (c) the change in parasite load and serological response over the follow-up period, as measured by quantitative PCR (qPCR); (d) the changes in biomarker levels at different points of the follow-up and the correlate with parasite load and ATO regimen; and € the incidence of treatment discontinuation due to severe AEs.

Clinical CD-related events occur at a low incidence in indeterminate CD patients because of the pathogenesis and hysteresis of the disease, parasite characteristics, and host response times; thus, it would require extensive clinical trials to assess incidence changes. Also, serological and clinical end-points of cure would need several years to decades in chronic CD [6]. Consequently, the criterion of cure as a primary outcome, as determined by seroreversion, is not considered in this proposal.

**Study description.**    This is a proof-of-concept phase II clinical trial with three different arms: two arms will receive 40 and 80 mg/day ATO, respectively. The third arm will receive a placebo. All arms will receive antiparasitic therapy before ATO. ATO doses were chosen to minimize false-negative results, thus, providing the best test of the hypothesis and maximizing the pharmacodynamic effect on inflammation and cardiac and endothelial function. Besides, separating antichagasic drugs and ATO will minimize the probability of severe drug interactions, primarily because of the risk of hepatotoxicity. Study subjects will be randomized, and all three arms will be double-blinded for ATO or placebo. However, antichagasic therapy will be open-labeled. Also, qPCR, other laboratory assessments, ECGs, and 2D echocardiograms will be performed with the clinical investigators blinded to ATO or placebo allocations.

This clinical trial will be conducted in four Chilean centers: 1) Hospital San Juan de Dios and 2) Hospital Felix Bulnes in Santiago, Región Metropolitana, 3) Hospital Dr. Gustavo Fricke in Viña del Mar, and 4) Hospital Interprovincial Quillota-Petorca, Región de Valparaíso. Well-established Programs for Chagas Control (PCC) are ongoing in all those centers. Moreover, each center incorporates an average of 60 newly diagnosed patients yearly, allowing the recruitment of a sufficient number of subjects to achieve adequate statistical power to avoid type I and II errors within the sample size calculated for this type of study, as detailed below.

**Study population.**    Patients older than 18 and younger than 50 years old with chronic indeterminate CD attending the PCC will be enrolled equally and randomly into the four study arms. According to PCC guidelines, patients older than 50 are not eligible for antichagasic therapy. The choice of this CD target population is primarily driven by the unmet medical need for a new, safe, and effective treatment for chronic indeterminate CD.

**Study duration and duration of subject participation.**    The total duration of patient participation in the study will be twelve months, including two weeks for screening and evaluation according to the PCC guidelines and eight weeks of antichagasic treatment. After two weeks of washout, the administration of ATO and placebo will continue for a further eight-week period. There will be follow-up visits up to twelve months after treatment initiation. The SPIRIT schedule is depicted in **Fig 1**.

Patients will initiate a 14-day evaluation phase after written, voluntary informed consent. Once a patient is randomized, and the treatments are started, s/he will have follow-up visits during the treatment phase of the study on Days 30, 60, and 120 (allowable window of ± four days), and three visits after the end of treatment (EOT) at Day 180, 240, and 360 (allowable window of ± 14 days). Also, patients will be advised to return on any day during the follow-up period if they present any medical occurrence or AE. A graphical description of this timeline is included in **Fig 2**.

| Phase | Pre-Randomization | | Post-Randomization | | | | | |
|---|---|---|---|---|---|---|---|---|
| Period | Screening | baseline | treatment | | | | Follow-up | |
| Visit | 1 | 2 | 3 | 4 | 5 | 6 | 7 | 8 |
| Day(s) | -14 a -1 | 0 | 30 | 60 | 120 | 180 | 240 | 360 |
| **Procedure** | | | | | | | | |
| Informed Consent | x | | | | | | | |
| Randomization | | x | | | | | | |
| Clinical Record | x | | | | | | | |
| Previous/concurrent drugs | | ←----------------------------------------------------→ | | | | | | |
| Inclusion/Exclusion | x | x | | | | | | |
| Vital Signs | x | x | x | x | x | x | x | x |
| Complete physical exam | x | | | | x | | | x |
| Chest X-ray | x | | | | x | | | x |
| ECG | x | | | x | x | (x)e | (x)e | x |
| 2D Echocardiogram | x | | | | | | | x |
| Pregnancy test | x | | | | | x | x | |
| Serology | x | | | | | | | x |
| Laboratory | x | x | x | x | x | x | x | x |
| Quantitative PCR | x | | | | Xd | | | x |
| Biomarkersb | | X | | x | x | | | x |
| Focused Physical Examination c | x | x | x | x | x | x | x | x |
| Adverse Events | | | ←------------------------------→ | | | | | |
| Drug Accounting | | x | x | x | x | | | |

aLaboratory parameters will include: hemoglobin, total white blood cell count, differential white blood cell count, and platelet count. Laboratory biochemical parameters will include: CK, ALT, AST, GGT, alkaline phosphatase, total and direct bilirubin, a lipid profile: total cholesterol, c-HDL, and c-LDL, triglycerides, fasting blood glucose, and creatinine;
bBiomarkers: BNP, cTnT, IFN-γ, IL-1β, IL-4, IL-17A, e IL-10, sICAM-1, sVCAM-1, sE-selectin;
cPhysical examination focused only on the evaluation of adverse events.;
dPCR testing at these time points will be done with a single 10 ml sample;
eECGs will be performed at these visits only in case of previously identified abnormalities

**Fig 1. Schedule of events and procedures for each subject during the recruitment phase under the SPIRIT guidelines.**

Recruitment is expected to be completed within 24 months of recruitment commencement. Therefore, the timeline for First Patient In (FPI) and Last Patient Out (LPO) is 36 months. However, the total study duration is estimated to be 48 months, from the start-up phase to the final study report.

Inclusion and exclusion criteria for subject recruitment are defined in **Fig 3**.

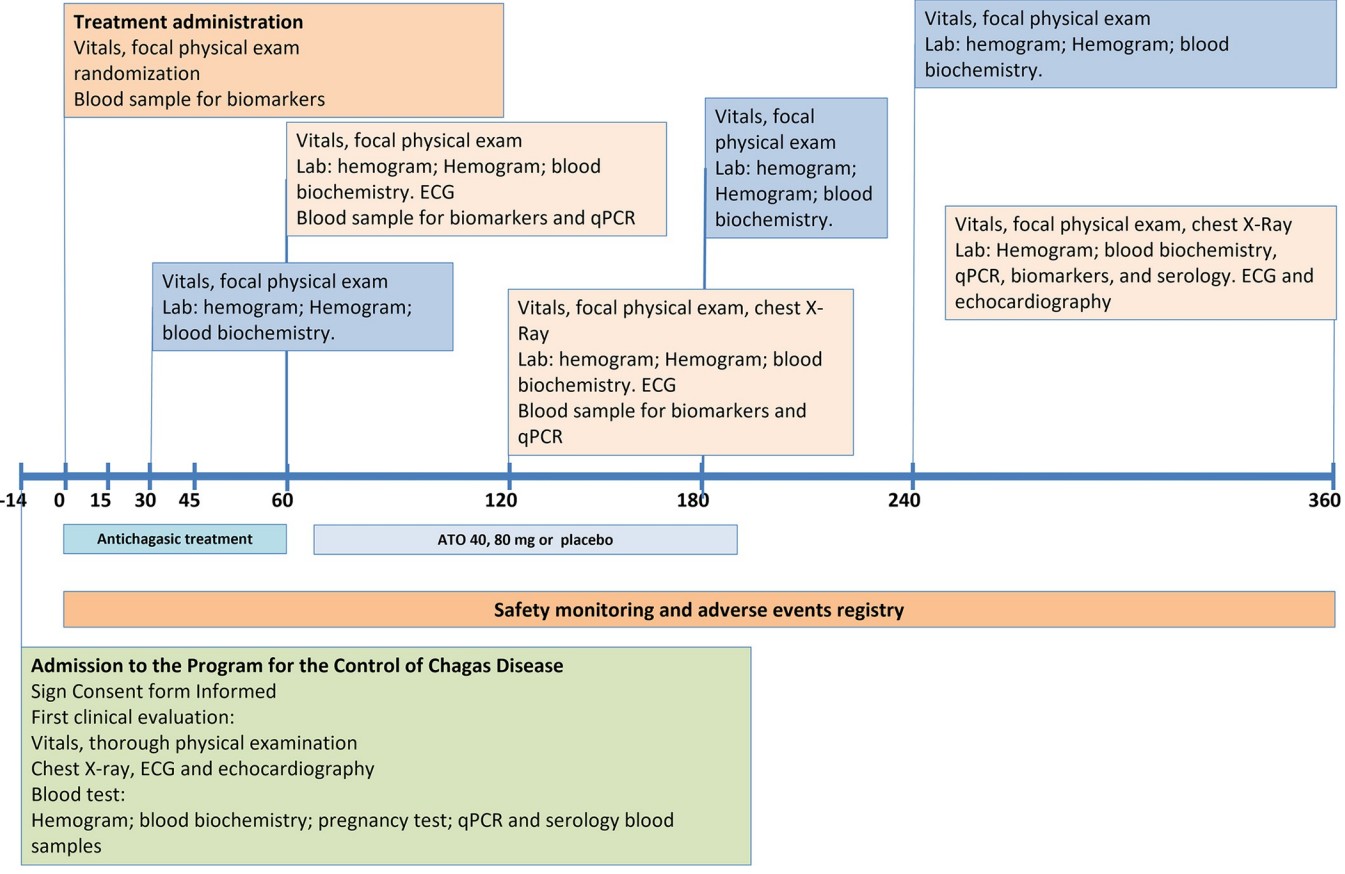

**Fig 2. Graphic timeline of the main events in the trial.**

Patients fulfilling the selection criteria will go through the randomization procedure. Only subjects enrolled in this trial will receive study medications. Patients will be assigned, in ascending order, a trial identification number (TIN) according to the chronological order of recruitment. A patient is considered randomized when s/he receives a TIN.

## Doses and treatment regimens

40 and 80 mg ATO will be purchased from a pharmaceutical laboratory recognized by the ISPCH that offers the active ingredient in a bioequivalent formulation.

The antichagasic drugs NFX or BZD will be supplied by Chilean Health Ministry as part of the PCC. Antichagasic therapy will be administered as directed by the " Manual of Procedures for the Care of Patients with Chagas Disease," [1] and ATO, antichagasic drugs, and placebo treatment administration will be supervised at each of the patient visits to determine pill accountability and drug compliance. BZD dose will be 5 mg/kg/day every 12 hours (BID), and NFX 5–10 mg/kg/day BID. ATO doses will be 40 and 80 mg/day for each respective arm and administered in the morning.

The patient will receive enough medication at each visit until the next scheduled visit. Also, they must bring all remaining study drugs on Day 30 and Day 60 visits to check for compliance with prescribed treatment and drug accountability. If a patient develops symptoms or signs of NFX intolerance, s/he can be switched to BZD without being withdrawn from the study.

| Inclusion criteria | Exclusion criteria | Withdrawal criteria |
|---|---|---|
| Adults older than 18 and younger than 50 years. | Signs and symptoms of the digestive form of CD. | Severe skin reactions or repetitive moderate skin reactions. |
| Body weight higher than 40 kg. | Chronic cardiac CD stage II or higher. | |
| conventional confirmatory serology for *T. Cruzi* infection from the Chilean Public Health Institute. | Acute or chronic health conditions such as acute infections, history of HIV infection, diabetes, liver, or renal disease. | Serum ALT, AST, GGT, CK exceeding 2 X ULN at any time |
| Positive qPCR. | Pre-existing heart disease not related to CD. | Elevation of serum glucose or bilirubin >2 X ULN at any time. |
| Ability to comply with all protocol-specified follow-up tests and visits. | Formal contraindication to receive NFX or BZD, | Serious adverse events. |
| Have a permanent address. | Known history of hypersensitivity, allergic, or severe adverse reactions to ATO, BZD, or NFX. | Any condition that the clinical investigator or any other attending physician not involved in the study consider medically necessary to interrupt treatment and withdraw a patient from the study. |
| Signed the written informed consent form; | A history of previous treatment for CD. | |
| | History of previous treatment with atorvastatin, lovastatin, rosuvastatin, simvastatin or any other statin. | Significant protocol deviation. |
| | Any concomitant use of antimicrobial agents. | Lost to follow-up. |
| | History of alcohol or drug abuse. | Patient/caregiver/legal representative withdrawal of the informed consent. |
| | Any condition precluding oral medications. | Study termination by the investigator |
| | Concomitant or anticipated use of CYP3A4 modifiers. | |
| | Medical history of familial short QT syndrome or concomitant therapy with medications that can shorten the qt interval. | |
| | Abnormal laboratory test values for the following parameters: | |
| | Total white blood cell count, | |
| | Platelet count, | |
| | ALT, AST, CK, total bilirubin, or creatinine, or GGT > 2 X ULN | |
| Women in reproductive age must have a negative serum pregnancy test, must not be breastfeeding, and must consistently use a highly effective contraceptive method during the entire treatment phase. | | |

Chagas disease (CD), Nifurtimox (NFX), Benznidazole (BZD), Atorvastatin (ATO), alanine aminotransferase (ALT), Aspartate transaminase (AST), γ-glutamyl transferase (GGT); creatine kinase (CK), Upper Limit of Normal (ULN).

**Fig 3. Criteria for inclusion, exclusion, and withdrawal of study subjects.**

## Adverse events and withdrawal from the study

An adverse event (AE) will be defined as any untoward medical occurrence (any unfavorable and unintended symptom, sign, or disease, including an abnormal laboratory or ECG finding) or the worsening of any pre-existing condition occurring during the study whether or not considered to be causally related to the study or study drugs. Abnormal laboratory (hematology and biochemistry) results will be reported as AE if:

a) Occurs or worsens after the start of the study treatments,

b) It is considered a clinically significant adverse change by the clinical investigator, or

c) Is higher than Common Terminology Criteria for AE Grade 1 [57] unless they are associated with an already reported clinical AE.

The clinical investigators, or appropriate site personnel (not involved in the study), will examine any subject experiencing an AE as soon as possible. The investigator will do whatever is medically necessary for the safety and well-being of the subject. The subject will remain under observation if s/he is receiving any trial drugs and for two months following the last day

of drug administration, or longer if medically indicated in the investigator's opinion. All AE observed or reported following administration of the investigational treatments would be followed until resolved or until medically stable. For this trial, the clinical investigators of each site will report to the principal investigator, who will be responsible for reporting AE to the Ethical Committees and the drug surveillance authority in the ISPCH. The clinical investigators should report all AEs directly observed and all spontaneously reported by the subjects, using concise medical terminology in the case report form (CRF). Also, during each trial visit, the subjects will be interviewed and undergo a focused physical exam for AE evaluation.

Patients will be considered to have withdrawn from the study if they had entered into the study (i.e., gave informed consent and received at least one dose of treatment) but did not complete the treatment phase of the study and the follow-up assessments after EOT. Other withdrawal criteria are indicated in **Fig 3**.

If a subject does not return for a scheduled visit, every effort should be made to contact the subject. In any circumstance, every effort should be made to document the subject outcome, if possible. If the subject withdraws consent, no further evaluations should be performed, and no attempts should be made to collect additional data, except for safety data, which should be collected if possible. Data obtained from withdrawn patients before his/her withdrawal will still be considered. The subjects withdrawn from this study will not be replaced.

Treatment discontinuation does not imply withdrawal from the study. In such cases, the treatment might be discontinued for a few days; therefore, treatment will be considered incomplete or delayed. Antichagasic drugs and ATO can be resumed according to the assessment of the clinical investigator in charge of the patient. These patients should continue with study visits and assessments as planned.

## Unblinding procedure

Codebreaking will be performed only in the rare event of a medical emergency or serious and unexpected AEs, when the physician in charge of the patient (and not the investigator responsible for this project) feels that the patient cannot be treated adequately unless the treatment allocation is known. A proper code-break report should be raised along with filling the AE Notification Form, as provided by the drug-surveillance system of the ISPCH.

Patients may receive concomitant therapy for medical occurrences during the study. All concomitant treatment taken by the patient during the study, from the date of signature of the informed consent until the last follow up visit, will be recorded in the appropriate section of the CRF.

## Assessments

Upon admission to the PCC, patients will undergo a complete medical history, with emphasis on CD; inquiring for demographic data and history of medications; Physical examination, body weight and height, vital signs, and body temperature.

10 ml of blood, in separate tubes, will be collected for hematologic and biochemical assessments: hemoglobin, total and differential white blood cell and platelet counts, ALT, AST, GGT, alkaline phosphatase, total and direct bilirubin, a lipid profile: total cholesterol, c-HDL and c-LDL, triglycerides, fasting glycemia, creatine kinase, and creatinine. Also, a serum pregnancy test will be performed. This test will be performed in the clinical laboratories of each center.

A 3 mL blood sample will be collected for conventional CD serology performed in the clinical laboratories at each center and for a confirmatory serology test that must be performed in the ISPCH.

5 mL of blood will be collected to perform the qPCR. For clinical development and proof-of-concept, it has been proposed that quantitative PCR parasitological tests be used as markers for efficacy in clinical studies of chronic indeterminate CD [18]. Thus, this trial would use the real-time PCR technique described by Duffy et al. [58]. Blood samples shall be immediately added to a tube containing one volume (10mL) of a solution of Guanidine/ClH 6M EDTA 0.2M pH 8,0 Buffer (GEB) [59]. Samples with guanidine buffer can remain at room temperature for 30 days. For more extended periods, they need to be stored in a refrigerator. After DNA extraction, with a commercial Kit, samples will be processed on an Applied 155 Biosystems 7300 RT-PCR system (Thermo Fisher).

Additional 3 ml blood will be drawn for biomarkers: BNP, cTnT, IFN-γ, IL-1β, IL-4, IL-17A, and IL-10, sICAM-1, sVCAM-1, sE-selectin. These markers were selected following the literature review [15, 16, 33]. The serum levels of these markers will be determined using a multiplex approach with a Luminex 200 (R&Dsystems). A pilot study will be conducted before analyzing the trial samples to establish biological reproducibility.

Finally, a baseline resting ECG and a 2D echocardiogram will be performed. Both exams will be performed at each center. ECG must be regular or with nonspecific changes (incomplete right bundle branch block, incomplete left anterior fascicular block, mild bradycardia, minor PR interval increase, and minor ST-T changes). Any clinically significant abnormalities found on the electrocardiogram will automatically lead to patient exclusion from this study (see exclusion criteria).

After randomization and before initiating treatments, patients will undergo a new focused physical examination, and blood will be drawn for qPCR and biomarker determination.

## Data analysis and statistical methods

**Sample size.** A sample size of 75 patients per arm is sufficient to reach a 90% power at a global 5% significance level (2-sided), assuming the proportion of patients that improve cardiac damage and function at EOT in the experimental treatment group is 0.40. The proportion in the placebo group (Pp-Placebo) is 0.20, with an odds ratio of 2.5. Considering an estimated drop-out rate of 25%, then 100 patients per group would be recruited (300 patients in total) [60]. The study will be sufficiently powered to provide evidence of superior efficacy of either dose of ATO plus antichagasic therapy relative to antichagasic treatment alone.

**Randomization and treatment allocation.** A computer-generated randomization list will be prepared and stratified by center using a block size of twelve. Each center will receive a list of randomization numbers and the corresponding ATO or placebo packages. After the patient enrolls, the following randomization number in the respective center (in chronological order) will be assigned, and the corresponding treatment package will be delivered.

Populations to be analyzed: Three populations will be included in the analysis: i) the intention-to-treat (ITT) population comprising all randomized patients by their assigned treatment arms, ii) the per-protocol population composed of all ITT patients without any significant protocol deviations, and iii) the safety population comprising all patients randomized and having received at least one dose of study therapy.

Descriptive statistics on the per-protocol and ITT populations will be presented for demographics and baseline characteristics. For the primary outcome analysis, a one-sided Fisher's exact test of the proportion of patients with a significant decrease in biomarker levels on the ITT (primary analysis), and per-protocol (secondary analysis) populations will be used. ANOVA test will be performed to analyze the differences in the variances among the different markers. An exact test will be conducted for all secondary comparisons of proportions between ATO versus placebo, including parasitological response to treatment. Latent class and

multivariate analyses will be used to evaluate the association between the parasitological response, changes in biomarkers, and ATO dose.

**Safety analysis.** The proportion of patients presenting at least one AE will be described. The incidence rate and 95% confidence interval will be delivered per study arm. Otherwise, only descriptive statistics will be presented. Safety laboratory parameters (hematology and biochemistry) will also be described individually per study arm, showing the proportion of patients by degree of elevation relative to ULN and to baseline values and blood level changes over time.

## Data management

The present clinical trial will use an electronic CRF hosted on the RedCap platform of the Faculty of Medicine of the University of Chile, which also allows the storage of study data while maintaining confidentiality by current national legislation on patient data.

## Bioethics

The trial protocol (**S2 File**) and informed consent form are approved by the Ethics Committee at the Faculty of Medicine of the University of Chile (Proyecto 009–2021; S4 File) and received the respective ISPCH authorization (Resolution N˚ 13443/22) by the law N˚ 20120, and its respective regulations. Written informed consent will be obtained from all participants for inclusion in the study.

## Discussion

There is a general agreement that adults with chronic indeterminate CD are the population with the most urgent requirements for developing new treatments because of the highest disease burden on these patients. Thus, improving the host's factors (e.g., the immune reaction elicited) may increase the efficacy of the conventional antichagasic therapy, probably by decreasing the dose, reducing its duration, or both.

There is only one mention of the use of rosuvastatin in treating CD, but the quality of the provided evidence is poor [61]. Other clinical trials that focused only on symptomatic treatment of heart failure associated with CD have assayed amiodarone [62], carvedilol [63], and angiotensin-modulating drugs [64]. Other drugs studied in small clinical trials are allopurinol, itraconazole, or ketoconazole, with mixed results [65–67]; unfortunately, more clinical trials with these drugs are lacking.

Given the sparse evidence-based treatment in this disease, this study will provide an opportunity to assess the impact of a novel therapeutic strategy on a combination of promising candidate markers of inflammation, endothelial activity, and cardiac function associated with sustained improvement and correlate findings with parasitological outcomes. Hence, this trial could be the first step to evaluate a potential therapy that it is proposed will favorably modify the course of chronic indeterminate CD.

## Supporting information

**S1 File. SPIRIT 2013 checklist.**
(DOCX)

**S2 File. Protocol approved by the Ethics Committee (English version).**
(PDF)

**S3 File. Protocol approved by the Ethics Committee (Spanish).**
(PDF)

**S4 File.**
(PDF)

## Acknowledgments

The authors thank the contribution of the health professionals Gabriel Llull from San Juan de Dios Hospital, Fernando Pasten from Felix Bulnes Hospital, Camila Nanjari and Dobrana Torres from the Quillota-Petorca Interprovincial Hospital, Sandra Rossi and Doris González from the Gustavo Fricke Hospital, and all the authorities of the participating centers for the support giving for the accomplishment of the research.

## Author Contributions

**Conceptualization:** Carolina Campos-Estrada, Edurne Urarte, Marisol Denegri, Litzi Villalón, Ulrike Kemmerling, Juan D. Maya.

**Funding acquisition:** Juan D. Maya.

**Investigation:** Carolina Campos-Estrada.

**Methodology:** Carolina Campos-Estrada, Edurne Urarte, Marisol Denegri, Litzi Villalón, Fabiola González-Herrera, Ulrike Kemmerling, Juan D. Maya.

**Project administration:** Juan D. Maya.

**Resources:** Fabiola González-Herrera.

**Supervision:** Edurne Urarte, Marisol Denegri.

**Validation:** Fabiola González-Herrera.

**Writing – original draft:** Juan D. Maya.

**Writing – review & editing:** Carolina Campos-Estrada, Litzi Villalón, Ulrike Kemmerling, Juan D. Maya.

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
