## [Decision Letter · Decision Letter 0]

31 Oct 2022

PONE-D-22-25692Effect of statins on inflammation and cardiac function in patients with chronic Chagas disease: A protocol for pathophysiological studies in a multicenter, placebo-controlled, proof-of-concept phase II trialPLOS ONE

Dear Dr. Maya,

Thank you for submitting your manuscript to PLOS ONE. After careful consideration, we feel that it has merit but does not fully meet PLOS ONE’s publication criteria as it currently stands. Therefore, we invite you to submit a revised version of the manuscript that addresses the points raised during the review process.

We look forward to receiving your revised manuscript.

Kind regards,

Yoshihiro Fukumoto

Academic Editor

PLOS ONE

Journal Requirements:

https://journals.asm.org/doi/full/10.1128/aac.02137-16

https://storage.googleapis.com/plos-corpus-prod/10.1371/journal.pntd.0004880/1/pntd.0004880.s002.pdf?X-Goog-Algorithm=GOOG4-RSA-SHA256&X-Goog-Credential=wombat-sa%40plos-prod.iam.gserviceaccount.com%2F20220920%2Fauto%2Fstorage%2Fgoog4_request&X-Goog-Date=20220920T050126Z&X-Goog-Expires=86400&X-Goog-SignedHeaders=host&X-Goog-Signature=47523d5c42db05cb348b1362665b3ae84f7a9685346a65c811f9d2f00b9b115897a4956c9ad853c93b87fe31836be02b567249e8975296493ad768f9d96c7997fcd413f3c03fb9d6aee3e60c0fdb8e349cf35f68bc76f64eb23c03881cdb036deaa48687d3499ebced53bf69d7e28cc795dd0d4551f944013f1feddcd7548fb20668b2731d9c87a8864aec932e22631065146cb9f71ba3f06d68e086a1c08cdfd3dea374e2fecd397929462e877e7b3266580f73515661f04eff8773a3f72853dadedc68c65c11c4efb85e0b99f803db0fdb5e813ace4a7d18b58218564bf45f543fe2f47fd76605bd3c51748cfff20a46a7e02d8062a7ef79a27624813c9431

Hence, I recommended it is ok to proceed with the PRTC note on minor text overlap.

In your revision ensure you cite all your sources (including your own works), and quote or rephrase any duplicated text outside the methods section. Further consideration is dependent on these concerns being addressed.

Reviewers' comments:

Reviewer's Responses to Questions

**Comments to the Author**

1. Does the manuscript provide a valid rationale for the proposed study, with clearly identified and justified research questions?

Reviewer #1: Yes

Reviewer #2: Yes

2. Is the protocol technically sound and planned in a manner that will lead to a meaningful outcome and allow testing the stated hypotheses?

Reviewer #1: Yes

Reviewer #2: Yes

3. Is the methodology feasible and described in sufficient detail to allow the work to be replicable?

Reviewer #1: Yes

Reviewer #2: Yes

4. Have the authors described where all data underlying the findings will be made available when the study is complete?

Reviewer #1: Yes

Reviewer #2: No

5. Is the manuscript presented in an intelligible fashion and written in standard English?

Reviewer #1: Yes

Reviewer #2: Yes

6. Review Comments to the Author

You may also provide optional suggestions and comments to authors that they might find helpful in planning their study.

Reviewer #1: This is an interesting protocol to assess effect of statins on inflammation and cardiac function in patients with chronic Chagas disease from a multicenter, placebo-controlled, proof-of-concept phase II trial. Well written and followed by the SPIRIT 2013 statement. I have a few comments to the authors.

1. The primary outcome of this study is the change in cardiac function, but sample size calculation was done by change of inflammation biomarkers. Why not the authors conduct sample size calculation by the primary outcome? Could the authors comment on it?

2. Please check figure and table numbers throughout manuscript. E.g. page 12. line 247, "Inclusion and exclusion criteria for subject recruitment are defined in figure 2". Page 14, line 293-4 "Other withdrawal criteria are indicated in Fig. 2." These should be Figure 3.

Reviewer #2: I have reviewed the study protocol paper entitled “Effect of statins on inflammation and cardiac function in patients with chronic Chagas disease: A protocol for pathophysiological studies in a multicenter, placebo-controlled, proof-of-concept phase II trial” with great interest.

This study will provide an important contribution to the treatment of Chagas disease.

The protocol has been developed well.

I feel this paper would be appropriate for the journal.

7. PLOS authors have the option to publish the peer review history of their article (what does this mean?). If published, this will include your full peer review and any attached files.

Reviewer #1: No

Reviewer #2: No

---

## [Author Response · Author response to Decision Letter 0]

5 Dec 2022

On behalf of all the authors of the present protocol, we would like to thank the reviewers for their thorough work, which undoubtedly improved the new version of the manuscript.

Below you will find the point-by-point responses to each indication or suggestion made by the reviewers. 

6. Review Comments to the Author

You may also provide optional suggestions and comments to authors that they might find helpful in planning their study.

Reviewer #1: This is an interesting protocol to assess effect of statins on inflammation and cardiac function in patients with chronic Chagas disease from a multicenter, placebo-controlled, proof-of-concept phase II trial. Well written and followed by the SPIRIT 2013 statement. I have a few comments to the authors.

1. The primary outcome of this study is the change in cardiac function, but sample size calculation was done by change of inflammation biomarkers. Why not the authors conduct sample size calculation by the primary outcome? Could the authors comment on it?

We thank the reviewer for drawing attention to this point. A transcription issue was maintained from the other document versions that remained. Nonetheless, the sample size calculation was made considering the improvement in cardiac function. For this purpose, the composite analysis of the plasma levels of BNP, and cTnT, the resting 12-lead ECG (cardiac rate, QT segment duration, changes in electrical conduction as determined by QRS segment duration and morphology), and ejection fraction determined by a 2D-echocardiogram is proposed as an indicator. Thus, line 356-357 vas corrected, as can be seen in the marked version.

2. Please check figure and table numbers throughout manuscript. E.g. page 12. line 247, "Inclusion and exclusion criteria for subject recruitment are defined in figure 2". Page 14, line 293-4 "Other withdrawal criteria are indicated in Fig. 2." These should be Figure 3.

Thank you for this indication. The proper call for figure 3 was corrected. 

Reviewer #2: I have reviewed the study protocol paper entitled “Effect of statins on inflammation and cardiac function in patients with chronic Chagas disease: A protocol for pathophysiological studies in a multicenter, placebo-controlled, proof-of-concept phase II trial” with great interest.

This study will provide an important contribution to the treatment of Chagas disease.

The protocol has been developed well.

I feel this paper would be appropriate for the journal.

Again, thank you very much for your kind comments.

---

## [Decision Letter · Decision Letter 1]

27 Dec 2022

Effect of statins on inflammation and cardiac function in patients with chronic Chagas disease: A protocol for pathophysiological studies in a multicenter, placebo-controlled, proof-of-concept phase II trial

PONE-D-22-25692R1

Dear Dr. Maya,

We’re pleased to inform you that your manuscript has been judged scientifically suitable for publication and will be formally accepted for publication once it meets all outstanding technical requirements.

Kind regards,

Yoshihiro Fukumoto

Academic Editor

PLOS ONE

Additional Editor Comments (optional):

Reviewers' comments:

Reviewer's Responses to Questions

**Comments to the Author**

1. Does the manuscript provide a valid rationale for the proposed study, with clearly identified and justified research questions?

Reviewer #1: Yes

2. Is the protocol technically sound and planned in a manner that will lead to a meaningful outcome and allow testing the stated hypotheses?

Reviewer #1: Yes

3. Is the methodology feasible and described in sufficient detail to allow the work to be replicable?

Reviewer #1: Yes

4. Have the authors described where all data underlying the findings will be made available when the study is complete?

Reviewer #1: Yes

5. Is the manuscript presented in an intelligible fashion and written in standard English?

Reviewer #1: Yes

6. Review Comments to the Author

You may also provide optional suggestions and comments to authors that they might find helpful in planning their study.

Reviewer #1: The authors addressed my comments promptly. In this revised manuscript, I do not have further comments to the authors.

7. PLOS authors have the option to publish the peer review history of their article (what does this mean?). If published, this will include your full peer review and any attached files.

Reviewer #1: No

---

## [Editor Report · Acceptance letter]

4 Jan 2023

PONE-D-22-25692R1 

Effect of statins on inflammation and cardiac function in patients with chronic Chagas disease: A protocol for pathophysiological studies in a multicenter, placebo-controlled, proof-of-concept phase II trial 

Dear Dr. Maya:

I'm pleased to inform you that your manuscript has been deemed suitable for publication in PLOS ONE. Congratulations! Your manuscript is now with our production department. 

Kind regards, 

on behalf of

Dr. Yoshihiro Fukumoto 

Academic Editor

PLOS ONE